# Research on Diesel Engine Fault Status Identification Method Based on Synchro Squeezing S-Transform and Vision Transformer

**DOI:** 10.3390/s23146447

**Published:** 2023-07-16

**Authors:** Siyu Li, Zichang Liu, Yunbin Yan, Rongcai Wang, Enzhi Dong, Zhonghua Cheng

**Affiliations:** Shijiazhuang Campus of Army Engineering University of PLA, Shijiazhuang 050003, China; sy_li1988@163.com (S.L.); zc_liu1997@aeu.edu.cn (Z.L.); siyusi@126.com (Y.Y.); wrcpromising@aeu.edu.cn (R.W.); ez_dong@aeu.edu.cn (E.D.)

**Keywords:** synchro squeezing S-transform, vision transformer, diesel engine, fault status identification, reliability, time-frequency analysis

## Abstract

The reliability and safety of diesel engines gradually decrease with the increase in running time, leading to frequent failures. To address the problem that it is difficult for the traditional fault status identification methods to identify diesel engine faults accurately, a diesel engine fault status identification method based on synchro squeezing S-transform (SSST) and vision transformer (ViT) is proposed. This method can effectively combine the advantages of the SSST method in processing non-linear and non-smooth signals with the powerful image classification capability of ViT. The vibration signals reflecting the diesel engine status are collected by sensors. To solve the problems of low time-frequency resolution and weak energy aggregation in traditional signal time-frequency analysis methods, the SSST method is used to convert the vibration signals into two-dimensional time-frequency maps; the ViT model is used to extract time-frequency image features for training to achieve diesel engine status assessment. Pre-set fault experiments are carried out using the diesel engine condition monitoring experimental bench, and the proposed method is compared with three traditional methods, namely, ST-ViT, SSST-2DCNN and FFT spectrum-1DCNN. The experimental results show that the overall fault status identification accuracy in the public dataset and the actual laboratory data reaches 98.31% and 95.67%, respectively, providing a new idea for diesel engine fault status identification.

## 1. Introduction

Diesel engines, as the main power source for heavy vehicles, construction machinery, ships, generator sets, tanks, etc., will greatly affect normal production and safe operation when they fail [1]. However, due to the high pressure and temperature generated during their operation, requiring high structural strength and stiffness of each relevant part, their reliability and safety gradually decrease with the increase in operation time [2]. The important components of a diesel engine will lose their original functions as their functions decay. Therefore, the safety and durability of diesel engines are receiving more and more attention, and the fault status identification of diesel engines has become a hot spot for research in related fields at home and abroad [3].

When identifying the status of a diesel engine, the original signal collected by the sensor is mainly a vibration signal. Because the collection of vibration signals is simple and fast, there is no need to disassemble the diesel engine body and change the diesel engine structure. Usually, the basic idea of fault status identification for diesel engines is firstly, collecting one-dimensional vibration data during the engine working condition, secondly, completing noise reduction and feature extraction of the original signal, and finally, completing fault status identification through a pattern recognition method [4].

Compared with traditional machine learning methods, deep learning has a powerful adaptive feature-learning capability to independently build the desired network model based on the sample data during the learning process, and has received much attention in the field of prognostics and health management [5]. Although deep learning is capable of training one-dimensional raw signals, two-dimensional images contain richer feature information and have received attention in this field as inputs for deep learning [6]. Compared with one-dimensional signals, two-dimensional images have two main advantages in fault status recognition: first, image data can consider multiple dimensions simultaneously when expressing information, whereas one-dimensional data can only consider one dimension of information, which is more one-sided [7]; secondly, images are easy to identify and classify, and the use of advanced algorithms to convert one-dimensional signals into two-dimensional images makes it easier to classify images from a visual perspective. Therefore, image data can provide more comprehensive and rich information for a wider range of application scenarios, such as image recognition, video analysis, and medical image processing [8]. Traditional time-frequency transformation methods, including short-time Fourier transform (STFT), continuous wavelet transform (CWT), S-transform (ST), etc., have achieved good results in studying time-frequency analysis [9]. After performing the time-frequency transform, the resulting feature maps are preprocessed and then input to a deep learning model for status recognition [10]. Liu et al. [11] converted diesel engine cylinder head vibration signals into time-frequency maps by STFT, which were input to an AlexNet network and ResNet-18 network for training, and achieved good fault classification by transfer learning algorithm. Xi et al. [12] used ST to convert diesel engine vibration signals into time-frequency maps and t-SNE to visualize fault features, which were input to an extreme learning machine classifier to intelligently classify diesel engine faults. Shen et al. [13] performed the Gabor transform on the vibration signal to obtain the time-frequency diagram of each operating status of the diesel engine. Mou et al. [14] converted the diesel engine vibration signal into a time-frequency map by smoothing the pseudo-Wigner distribution. However, the above method has problems such as fixed time-frequency resolution, little phase information, low resolution and poor energy aggregation [15]. The limited feature information contained in the conversion of the original vibration signal into an image by the above methods makes it difficult to effectively extract the time-dependent features of the vibration signals of the different statuses of the components to be monitored, which easily causes the loss of useful information and affects the recognition accuracy of the model [16]. Synchro squeezing wavelet transform (SSWT) is a combination of a synchro squeezing algorithm to squeeze the energy in a certain range around the center frequency of each frequency band to converge to the center frequency after wavelet transform to improve the resolution. Compared with the wavelet transform, the S-transform has better adaptivity and time-frequency resolution. The synchro squeezing S-transform (SSST), to a certain extent, solves the problems of poor adaptivity of SSWT and the low resolution of high-frequency low-amplitude signals, and has good effects in the time-frequency analysis of seismic signals and vibration signals.

In terms of pattern recognition technology, the convolutional neural network (CNN) is usually used to identify diesel engine fault status. Chen [17] established a diesel engine overall status identification model based on a support vector machine (SVM). However, SVM has the problem of being sensitive to parameter selection, which limits the application of the method in diesel engine condition identification. To solve the problems of index selection and the difficulty of weight determination in the traditional fault status identification method, Bai et al. [18] constructed a diesel engine fault status identification model based on CNN. Jiang et al. [19] addresses the problem that diesel engine faults are difficult to identify accurately under complex operating conditions, and the diesel engine vibration signals are fed into a one-dimensional CNN and a deep neural network of a long short-term network (LSTM) for training, which can be effective for status identification. Zhan et al. [20] proposed a fault identification method based on the combination of optimal variational mode decomposition (VMD) and an improved CNN. However, when classifying and recognizing images, the initial status parameters for the CNN can have a great impact on the network training, and a poor choice can cause the network to not work or potentially fall into local minima, underfitting, and overfitting. In 2019 researchers started to try to apply transformers to the CV domain, and finally in 2021, those involved proved that transformers have better scalability than CNNs, can handle sequential types of inputs, and are significantly better than CNNs when training larger models on larger datasets [21]. Alexey et al., proposed the vision transformer (ViT) model by directly applying the transformer architecture to image classification tasks, representing the input image as a feature vector that can be used for subsequent tasks; ViT significantly improves the performance of traditional image classification tasks [22].

Therefore, this paper takes diesel engines as the engineering research background, and proposes a diesel engine status recognition method that combines the advantages of SSST to represent time-varying nonlinear non-smooth signals with the excellent image classification ability of ViT to achieve the classification of diesel engine fault status in response to the current problem of inaccurate diesel engine status recognition. The main contributions and innovations of this paper are as follows:(1)Relying on the existing conditions in the laboratory, a pre-set fault experiment was carried out to realize the acquisition of diesel engine cylinder head vibration signals.(2)The original diesel engine vibration signal is represented as a time-frequency image by SSST, and the dependence of the vibration signal on time is mapped into the image feature space, so that the original feature information is retained in the time-frequency map as much as possible. Then, after applying the powerful learning ability of ViT to automatically extract the temporal and spatial features in the images, the fault status identification is completed.(3)The feasibility and effectiveness of the proposed diesel engine status recognition method is verified by means of public datasets and actual laboratory measurements.

The remaining sections are as follows: Section 2 introduces the relevant theories of SSST-ViT in detail; Section 3 provides the diesel engine pre-set fault experiments and the experimental data acquisition method, and the experimental results are analyzed and studied; In Section 4, the conclusions of this study and the outlook for future research work are presented.

## 2. Diesel Engine Fault Status Identification Method

In the diesel engine fault status identification method based on SSST-ViT, the original vibration signals collected are converted into time-frequency maps by the SSST, and the ViT network model is applied to identify each fault status. Therefore, in this section, SSST, the ViT network model and a diesel engine fault status identification method based on SSST-ViT are introduced.

### 2.1. Synchro Squeezing S-Transform

The ST of the acquired original diesel engine vibration signal is compressed synchronously and represented as a two-dimensional time-frequency map. Compared with CWT, ST has a better time-frequency discrimination effect. The result of the wavelet transform is a time-scale spectrum, and the result of ST is time-frequency spectrum, which is more intuitive and clearer, and is a reversible transform without signal loss [23]. In particular, it has an enhancement effect on the high-frequency weak amplitude components of the original signal, which is effective for weak signal testing and research analysis. The raw diesel engine signal collected by the vibration acceleration sensor is a typical one-dimensional time series, whose vertical coordinate is the amplitude corresponding to each sampling point and the horizontal coordinate is the time or sampling point. The original vibration signal cannot fully represent the fault status information of the diesel engine, and in order to effectively characterize the time-frequency characteristics of the original signal, the signal is converted into a time-frequency map, which can not only highlight the original characteristic information of the vibration signal, but can also further enhance the characteristic time series information [24].

Based on the principle of synchro squeezing transformation, the derivation process of SSST is as follows:

Define the ST equation of the signal x(t) as
(1)$${\mathrm{ST}}_{X}(f,b)=\int_{-\infty }^{+\infty }x(t)\frac{\left|f\right|}{\sqrt{2\pi }}e^{\frac{-f^{2}{\left(t-b\right)}^{2}}{2}}e^{-i2\pi ft}dt$$
where S(f,b) is the time-frequency spectrum of x(t), t is the time, f is the frequency, b is the displacement parameter, and i is the imaginary number.

$\varphi (t)=\frac{1}{\sqrt{2\pi }}e^{\frac{t^{2}}{2}}e^{i2\pi t}$, then Equation (1) yields
(2)$${\mathrm{ST}}_{X}(f,b)=\left|f\right|e^{-i2\pi fb}\int_{-\infty }^{+\infty }x(t)\bar{\varphi \left[f(t-b)\right]}dt$$

$\bar{\varphi }$ and φ are complex conjugates, and by Parseval’s theorem and Fourier transformations
(3)$${\mathrm{ST}}_{X}(f,b)=\frac{1}{2\pi }e^{-i2\pi fb}\int_{-\infty }^{+\infty }\hat{x}(\xi )\bar{\hat{\varphi }\left[(f^{-1}\xi )e^{ib\xi }\right]}d\xi$$
where ξ is the angular frequency, x(t) is obtained by Fourier transforming $\hat{x}(\xi )$, $\bar{\hat{\varphi }(\xi )}$ is the complex conjugate of φ(t), and the single harmonic case: $x(t)=A\cos(2\pi f_{0}t)$
(4)$$\hat{x}(\xi )=A\pi \left[\delta (\xi -2\pi f_{0})+\delta (\xi +2\pi f_{0})\right]$$

S-transform is
(5)$${\mathrm{ST}}_{X}(f,b)=\frac{A}{2\pi }e^{-i2\pi {\left(f-f_{0}\right)}^{b}\hat{\varphi }*(2\pi f^{-1}f_{0})}$$

The analysis shows that the energy of the time spectrum of x(t) is distributed at $f=f_{0}$, but the actual time spectrum is around $f_{0}$ with a spurious spectral bandwidth. The goal of SSST is to obtain the real instantaneous frequency of x(t) by converging the energy after compression.

Derivative of (5) with respect to time:(6)$$\frac{\partial }{\partial b}{\mathrm{ST}}_{X}(f,b)=-i\pi A(f-f_{0})e^{-i2\pi {\left(f-f_{0}\right)}^{b}}\bar{\hat{\varphi }(2\pi f^{-1}f_{0})}$$

The derivative of the time-frequency spectrum yields $\hat{f}$
(7)$$\hat{f}=(f,b)=f+{\left[i2{\pi \mathrm{ST}}_{x}(f,b)\right]}^{-1}\frac{\partial }{\partial b}{\mathrm{ST}}_{X}(f,b)$$

According to $x(t)=A\cos(2\pi f_{0}t)$, from (7) we can calculate
(8)$$\hat{f}=(f,b)=f+{\left[i2\pi ST_{x}(f,b)\right]}^{-1}\frac{\partial }{\partial b}{\mathrm{ST}}_{X}(f,b)=f+\frac{i\pi A(f-f_{0})e^{-i2\pi {\left(f-f_{0}\right)}^{b}}\bar{\hat{\varphi }(2\pi f^{-1}f_{0})}}{i2\pi \frac{A}{2}e^{-i2\pi {\left(f-f_{0}\right)}^{b}\hat{\varphi }*(2\pi f^{-1}f_{0})}}=f_{0}$$

The spectrum in the interval $\left[{\hat{f}}_{c}-\frac{△}{2}{\hat{f}}_{c},{\hat{f}}_{c}+\frac{△}{2}{\hat{f}}_{c}\right]$ near the center frequency ${\hat{f}}_{c}$ is superimposed to obtain the simultaneous compression transform $\mathrm{SSST}({\hat{f}}_{c},b)$, which improves the resolution of the spectrum, and the expression is
(9)$${\mathrm{SSST}}_{x}({\hat{f}}_{c},b)={\left(△\hat{f}\right)}^{-1}\sum_{f_{j}:\left|{\hat{f}}_{c}(f_{j},b)-{\hat{f}}_{c}\right|\leq \frac{△{\hat{f}}_{c}}{2}}\left|{\mathrm{ST}}_{X}(f,b)\right|f_{j}△f_{j}$$
where $f_{j}$ is the discrete frequency of ST, $△f_{j}=f_{j}-f_{j-1}$ interval, ${\hat{f}}_{c}$ and $△{\hat{f}}_{c}$ are the center frequency and spectral bandwidth of the compressed interval. $△{\hat{f}}_{c}={\hat{f}}_{c}-{\hat{f}}_{c-1}$.

SSST is a loss-free invertible transform, ${\mathrm{SSST}}_{x}({\hat{f}}_{c},b)$ can be expressed as x(b), and its inverse transformation equation is
(10)x(b)=Re(CφCϕ)−1∑CSSSTX(f^c,b)△f^cCφ=0.5∫0∞−φ(ξ)ξ−1dξ¯,Cϕ=e−i2πfb+ϕ(f,b)f2

The time and frequency distributions from SSST are linear, and the original signal x(t) can be calculated from the results of the synchronous compression transform by inverting the above equation.

### 2.2. Vision Transformer Network Model

The ViT network model was presented at ICLR2021, and the model consists of three modules including Embedding layer, Transformer Encoder, and MLP Head (which is eventually used for classification.) The Transformer model is based entirely on a self-attentive mechanism without any convolutional or recurrent neural network layers and is not subject to local interaction limitations [25]. ViT was the first Transformer model used to replace CNNs and applied to image classification [26]. Although Transformer was originally applied to sequence-to-sequence learning on text data, it has now been extended to various modern deep learning in areas such as vision, target detection and image segmentation [27].

Embedding layer: For the standard Transformer module, the input is required to be a sequence of tokens (vectors), i.e., a two-dimensional matrix; for image data, the data format is a three-dimensional matrix $\left[H,W,C\right]$, not what the Transformer wants. So, we need to carry out a transformation of the data using an Embedding layer, firstly dividing an image into a bunch of patches of a given size, and secondly mapping each patch into a one-dimensional vector by a linear mapping. This is achieved directly through a convolutional layer that flattens out the two dimensions, turning it into a two-dimensional matrix, which is what the Transformer wants. Before being input to the transformer encoder, a class token and Position Embedding need to be added, and a class token is inserted into the resulting pile of tokens specifically for classification. 

Transformer Encoder: The main part is to normalize each token with layer norm, so as to simulate the whole sample data distribution. 

Multi-head self-attention mechanism: This allows the model to focus on the information at different positions and complete the interaction information between sequences.

The structure of MLP consists mainly of a fully connected layer and an activation function. The output weights of the multi-headed self-attentive mechanism are received and compared to identify the fault types. The structure of the ViT model is shown in Figure 1.

According to the ViT model structure diagram, a ViT block runs in the following steps:

Step 1: Take the image size 224×224 as an example: as the input, the image is divided into fixed patches with size 16×16, so the number of patches generated is 224×224/16×16=196, and the sequence of length 196 is obtained, the dimensions of patches are 16×16×3=768, the dimensions of linear projection layer are 768×N(N=768), so the dimension of the input after passing through the linear projection layer is still 196×768, that is, there are 196 tokens in total, and the dimension of each token is 768. There is also a class for classification, so the final dimension is 197×768. The image is converted into a sequence by the patch embedding layer.

Step 2: Positional encoding: ViT also needs to add positional encoding, which can be understood as a table with N rows; the size of N is the same as the length of the input sequence, each row represents a vector, and the dimension of the vector is the same as the dimension of the input sequence embedding (768). After adding the location encoding information, the dimension is still 197×768.

Step 3: LN/multi-head attention: The LN output dimension is still 197×768. With multi-head self-attention, if there is only one head, the dimension is 197×768. If there are 12 heads, 768/12=64, the dimensions are 197×64. There are 12 groups in total, and finally the output of the 12 groups is stitched together again, and the output dimensions are 197×768. Then in a layer of LN, the dimensions are still 197×768.

Step 4: MLP: The dimension is enlarged and then reduced back (197×768 enlarged to 197×3072, then reduced to 197×768). After this, the block dimensions are still the same as the input (197×768), so it is possible to stack multiple blocks. Special characters (cls) corresponding to the output as the encoder output represent the final image presentation, followed by an MLP for image classification. The formula is as follows.

Define the image x, $x_{}\in R^{H\times W\times C}$, C is the number of channels, divided into N blocks of P*P images, N=H*WP*P.
(11)$$Z_{0}=\left[x_{\mathrm{cls}};x_{p}^{1}E;x_{p}^{2}E;\cdots ;x_{p}^{N}E\right]+E_{pos}\;E\in R^{p^{2}\times c\times D},E_{pos}\in R^{p(N+1)\times D}$$

At layer l ($l\in \left[1,N-1\right]$), the output is
(12)$$Z_{l}^{\prime }=MSA(LN(Z_{l-1}))+Z_{l-1}$$
(13)$$Z_{l}^{}=MLP(LN(Z^{\prime }{}_{l}))+Z_{l}{}^{\prime }$$
(14)$$y=LN(Z_{L}^{0})$$

Among them, the multi-headed attention layer MSE completes the information interaction between image blocks. The classification of image *y* is completed by the MLP. After superimposing ViT Blocks several times, the result of fault status recognition is output.

### 2.3. Diesel Engine State Identification Based on SSST-ViT

The diesel engine status recognition method of SSST-ViT effectively integrates the advantages of SSST in characterizing time-varying nonlinear non-smooth signals with the excellent image recognition capability of ViT, which can achieve accurate and efficient state recognition. The model structure diagram is shown in Figure 2, and the specific steps are as follows:

Step 1: The vibration signal of the diesel engine cylinder head is collected by the vibration sensor to obtain the raw dataset required for the experiment.

Step 2: The collected diesel engine vibration signals are transformed by SSST to obtain the time-frequency map. After pre-processing the time-frequency map, the required feature sample data are obtained. The sample data are divided into training set, validation set and test set according to a 7:2:1 ratio. The training set is used for model training, the validation set is used for initial evaluation of the accuracy of the model, and the test set is used for evaluating the performance of the model, which is not involved in the training of the model.

Step 3: The training set is used as the model input to the ViT network model and trained to obtain the desired diesel engine status recognition model.

Step 4: Use the test set as the trained model input to perform fault status recognition of the diesel engine.

## 3. Experimental Results and Comparative Analysis

The feasibility and effectiveness of the diesel engine condition identification method with SSST-ViT were validated using publicly available datasets and measured data from Case Western Reserve University (CWRU). The experiments were conducted using Windows 11 with a 12th Gen Intel(R) Core (TM) i7-12700H 2.30 GHz processor, a GeForce RTX 3060 Laptop GPU, 16G of RAM, Anaconda3, Python 3.9.13, and MATLAB2021b software environment. MATLAB2021b; deep learning framework is PyTorch1.11.0.

### 3.1. CWRU Dataset to Verify the Feasibility of SSST-ViT Method

Both rolling bearing and diesel engine vibration signals are characterized by time-varying, nonlinear non-smoothness [28]. Therefore, the feasibility of SSST with the ViT method was verified using the publicly available CWRU bearing vibration signal dataset. According to the literature, the publicly available dataset was obtained through a bearing failure simulation experiment bench.

The experiments were conducted using a deep groove ball bearing, model SKF6205, with a single point of failure of the bearing machined with electrical discharge machining (EDM), and the vibration acceleration signal of the bearing was collected using an accelerometer. The specific data used were the drive-end bearing data with a sampling frequency of 48 kHz, an approximate motor speed of 1797 r/min, and a load of 0 hp. The bearing statuses include normal, inner ring failure, outer ring failure, and rolling element failure, and each failure state can be classified into three types according to the depth of cut: 0.1778 mm, 0.3556 mm and 0.5334 mm. The status data of 10 bearings selected in this experiment are shown in Table 1.

The time-domain analysis of the vibration signals in each status of the bearing was performed. The data length of each fault status was intercepted to 5120 sampling points, and the time-domain waveforms of the bearing in 10 statuses were obtained, as shown in Figure 3.

As seen in the time-domain waveform diagram of the vibration signal, the time-domain waveforms of the status fluctuate widely, making it difficult to carry out effective fault status identification [29]. The signal waveforms of different status types are complex and do not differ greatly, and the individual status cannot be identified directly by hand. Therefore, it is difficult to carry out rolling bearing fault status identification from time-domain signal waveform analysis alone, and a more effective intelligent identification method is needed [30].

The SSST-ViT method proposed in this paper was applied to identify each status of the bearings. From each bearing status data point, 300 samples were randomly taken, and each sample length was 1024 sampling points, so a total of 3000 samples were obtained. By dividing the training set, validation set and test set according to the ratio of 7:2:1, 2100 training samples, 600 validation samples and 300 test samples were obtained for the feasibility verification experiments of SSST-ViT state recognition methods.

The SSST was performed on the original vibration signal to obtain the time-frequency diagram. To avoid the influence on the classification results, the coordinate system, legend and blank part were set not to be displayed, and the time-frequency diagram of the first sample in each state after processing is shown in Figure 4.

The warm and cold colors in Figure 4 represent energy values, the warmer the color the greater the energy, reflecting the energy magnitude of the signal at each frequency; the horizontal and vertical axes indicate time and frequency, respectively, showing the change of signal frequency components with time. The energy of the time-frequency diagram of each state is more concentrated, with good time-frequency resolution, and the contained features are different, corresponding to different time-frequency diagrams, with the warm color part showing irregular block distribution. Although there are certain differences in expression, the similarity is high, and it is difficult to manually distinguish each fault status accurately. Therefore, each fault status was identified by the ViT network with a powerful image classification function.

The images were first set to not show the legend, coordinate system and blank parts. Then each time-frequency map was normalized to speed up the model convergence. Finally, without affecting the recognition rate, the grid was normalized and compressed to process the time-frequency map to improve the model training speed, and the image size was uniformly adjusted to 224×224×3.

After considering the network structure, computer hardware level and sample characteristics and size, the parameters of the ViT network during training were configured as follows: batch processing size of 16; learning rate of 1 × 10^−3^; weight decay of 1 × 10^−5^; number of iterations—100; input image size of 224×224; number of classification categories—10; optimizer—stochastic gradient descent; and loss function—cross-entropy loss function. The experimental results were extracted from the training log and plotted.

ViT is the first transformer model used to replace the CNN and applied to image classification, which is able to achieve the desired results in the field of image classification. Therefore, in this paper, a ViT model is applied to diesel engine fault status identification. Since the network model is more suitable for extracting feature information from high-dimensional data, it is necessary to convert the one-dimensional vibration signals of diesel engines into two-dimensional images by some method. The S-transform combines the advantages of the continuous wavelet transform and the short-time Fourier transform, which has higher noise robustness and time-frequency analysis accuracy. However, the energy at a certain moment in the time-frequency map obtained by this method is distributed in a wider bandwidth near the instantaneous frequency, which causes instantaneous frequency energy leakage, leading to problems such as frequency band mixing and lower time-frequency resolution. Therefore, the synchro squeezing transform is combined with the S-transform to obtain the synchro squeezing S-transform method, which can effectively improve the time-frequency aggregation, time-frequency resolution and noise robustness compared with the traditional time-frequency analysis method. Therefore, the diesel engine fault status identification method proposed in this paper is obtained: SSST-ViT. The reason why ST-ViT is used as a comparison method in this paper is to verify that the SSST method can better characterize the feature information in the original signal and better retain the useful information. The reason why SSST-2DCNN is used as a comparison method in this paper is to verify that the ViT model has a more powerful image classification capability compared with the traditional CNN model, and is more suitable for identifying each fault status of the diesel engine. The reason why the FFT spectrum-1DCNN model is used as a comparison method in this paper is to verify that the 2D signal can better take into account the correlation of the signal in the time series compared to the 1D signal, i.e., to verify that the transformation of the original signal into a 2D image is more effective compared to the direct input of the 1D signal into the network model. In summary, it is completely reasonable to use the ST-ViT model, SSST-2DCNN model, and FFT spectrum-1DCNN model as comparisons in this paper.

In Reference [31], the authors proposed an intelligent fault diagnosis model for rolling bearings based on ViT, and achieved good results. Therefore, in this paper, the ViT model is applied to diesel engine fault status identification for the first time, and the feasibility and effectiveness of the proposed method are verified. Since ViT is the first transformer model used to replace the CNN and applied to image classification, and the network model is more suitable for extracting feature information from high-dimensional data, this paper proposes a diesel engine fault status recognition method based on SSST and ViT. Therefore, the ST-ViT model is improved on the basis of Reference [31], through which the model can be used to verify the superiority of the SSST time-frequency analysis method. In Reference [32], the authors proposed a 2DCNN-based fault diagnosis method for diesel engines by importing short-time Fourier transform (STFT) time-frequency maps into the 2DCNN model for training. However, the time-frequency map obtained using the STFT method suffers from low time-frequency resolution and weak energy aggregation. Therefore, the comparison method combines the SSST method with high time-frequency resolution and time-frequency aggregation with the 2DCNN, i.e., the SSST-2DCNN model is an improvement on Reference [32]. In Reference [33], the authors proposed a 1DCNN-based fault diagnosis method for diesel engines, where the features in the vibration signal are extracted and then input to the 1DCNN model for training. Since the method proposed in Reference [33] inputs multiple vibration signal features into the 1DCNN model for training, it tends to generate redundancy, which leads to a reduction in model efficiency. Therefore, the FFT spectrum is used as the fault feature in the comparison method of this paper. This is because different fault statuses of diesel engines generate different frequencies of fault features, and the FFT spectrum can well reflect the fault features of diesel engines at different fault states. Therefore, the source of the FFT spectrum-1DCNN model is Reference [33].

The training results of this model are compared with the training results of ST-ViT, SSST-2DCNN, and FFT spectrum-1DCNN models. The loss values and accuracy results of the training and validation sets of each model were obtained, as shown in Figure 5. The fault status identification results after 100 iterations are shown in Table 2 (Model 1: SSST-ViT; Model 2: ST-ViT; Model 3: SSST-2DCNN; Model 4: FFT spectrum-1DCNN).

As seen in Figure 5 and Table 2, the different fault status identification models have converged after 100 iterations and all perform well on the CWRU public dataset. In terms of model accuracy and loss values, the SSST-ViT method proposed in this paper has the fastest convergence speed at iteration, the highest accuracy and the lowest loss values on both the training and validation sets, and the best performance compared to the other three methods. In terms of training stability, the method is optimal, and the accuracy and loss value curves are generally very stable, while the other three comparison methods all show different degrees of fluctuations. Therefore, compared with the comparison models, the SSST-ViT fault status identification method has better performance in terms of accuracy, loss value and stability, and the feasibility of the proposed fault status identification method has been verified.

The accuracy and confusion matrix of different fault status identification models obtained under the test set are shown in Table 3 and Figure 6, respectively.

From Figure 6 and Table 3, it can be found that the proposed method has the optimal fault identification effect compared with other methods and can effectively distinguish the easily confused bearing fault types. To verify the feature extraction capability of SSST-ViT methods, the output of the classification layer network of the ViT model was extracted as discriminative features, and the identification results of bearing fault status were visualized in three dimensions by the t-SNE nonlinear dimensionality reduction technique, which is applicable to the visualization of high-dimensional data. The original data of the training set, the original data of the test set, the feature data of the training set and the feature data of the test set were obtained, as shown in Figure 7.

In Figure 7, using the test set feature data as an example, since none of the methods proposed in this paper achieved 100% accuracy under the test set, there must have been some points that did not fall within a cluster. In other words, it is because some features are identified as features of other fault statuses that some feature points are not in a cluster and therefore the accuracy is not 100%.

### 3.2. The Validity of the SSST-ViT Method Is Verified by the Measured Data

In order to verify the effectiveness of the SSST-ViT diesel engine fault status identification methods, this study relies on the high-pressure common rail diesel engine experimental bench in the laboratory, taking a CA6DF3-20E3 diesel engine as the research object to collect the state monitoring information during the operation of the diesel engine in different fault modes and provide data support for the research of diesel engine fault status identification methods. The experimental bench can be divided into two parts: the diesel engine system and the data acquisition system. The panoramic view of the experimental bench is shown in Figure 8, and the sensor installation is shown in Figure 9.

By analyzing the composition structure and function of the diesel engine, and combining its typical failure modes in the process of use and maintenance, the pre-set failure experiment is carried out in the diesel engine condition-monitoring experimental bench (by artificially processing or replacing the faulty parts, the diesel engine components are pre-set to collect the data in the engine fault status and carry out research). Typical failure modes of diesel engines were set as shown in Table 4:

In the actual equipment maintenance and repair process, due to the complex and harsh working environment, diesel engine failures may often be due to multiple, concurrent faults rather than a single failure mode. Therefore, when presetting the failure modes, a single failure mode is preset on the one hand and three mixed failure modes are preset on the other. As shown in Figure 10, the cylinder misfire fault is simulated by disconnecting the cylinder ignition power line, and the air intake outer cover is added to simulate the air filter blockage fault.

The diesel engine cylinder head vibration signal is acquired with a sampling frequency of 20 kHz, a single sampling time of 12 s, and a sample sampling interval of 30 s. After data acquisition experiments, there are 300 sets of data for each failure mode, six channels of data for each set, and a single sampling data volume of 240,000. In order to avoid the errors brought about by the process of the engine from start-up to steady status, the first 20 sets of data for each status are selected, and the last 20 sets of data from the 5th channel of each status are selected as the sample data.

Due to the simplicity, intuitiveness and clear physical meaning of the time-domain signal, the time-domain analysis of the vibration signal in each status of the diesel engine was performed. The length of the data intercepted for each failure mode sample is 5000 sampling points, and the time-domain waveforms of the diesel engine in each status are obtained as shown in Figure 11.

The engine head vibration signal presents a nonlinear and non-smooth status, and there are complex noise disturbances generated by the environment and the comprehensive action of each component during operation, so it is difficult to identify the fault status. From the time-domain waveform diagram in Figure 11, it can be seen that the vibration signal waveforms under different fault modes are complex and have basically the same amplitude change range, and there is no obvious difference from the time-domain waveform amplitude, so it is difficult to manually identify each status directly, so it is difficult to achieve effective identification of multiple engine faults from time-domain signal waveform analysis alone, and more effective fault information extraction and intelligent identification methods are needed.

The SSST-ViT method was applied to identify each status of the above diesel engine. A total of 2100 samples were obtained by taking 300 samples from each status of the diesel engine data, each with a sample length of 5000 sampling points. By dividing the training and validation sets according to the ratio of 7:2:1, 1470 training samples, 420 validation samples and 210 test samples were obtained, i.e., each status sample data included 210 training samples and 60 validation samples and 30 test samples. Processing of raw vibration signals by was carried out using the SSST and represented as a two-dimensional color time-frequency diagram, and the coordinate system, legend and blank part were set not to be displayed to avoid the influence on the classification results. The time-frequency diagram of the first sample in each status of the diesel engine after processing is shown in Figure 12.

Although each status in Figure 12 has some different expressions, the similarity is high, and it is difficult to distinguish each fault status only by hand. Therefore, each fault status is identified by the ViT network with a powerful image classification function.

The images were first set to not show the legend, coordinate system and blank parts. Then each time-frequency map was normalized to speed up the model convergence. Finally, the grid normalization compressed the time-frequency maps without affecting the recognition rate, and the image size was uniformly adjusted to 224×224×3.

After considering the network structure, computer hardware level and sample characteristics and size, the parameters of the ViT network during training were configured as follows: batch processing size of 16; learning rate of 1 × 10^−3^; weight decay of 1 × 10^−5^; discard rate of 0.1; number of iterations—100; input image size of 224 × 224; number of classification categories—7; optimizer—stochastic gradient descent; loss function—cross entropy loss function. The experimental results are extracted from the training log and plotted.

The training results of this model are compared with those of the ST-Vision Trans-former, SSST-2DCNN, and FFT spectrum-1DCNN models. The loss values and accuracy results of the training and validation sets of each model were obtained as shown in Figure 13. The fault status identification results after 100 iterations are shown in Table 5 (Model 1: SSST-ViT; Model 2: ST-ViT; Model 3: SSST-2DCNN; Model 4: FFT spectrum-1DCNN).

From Figure 13 and Table 5, it can be seen that in terms of model accuracy and loss values, the proposed SSST-ViT methods has the highest accuracy and lowest loss values in both training and validation sets with the best performance in terms of fast convergence during iterations compared with the other three compared methods. In terms of training stability, the accuracy and loss value curves of SSST-ViT methods is generally more stable. Therefore, compared with the comparison methods, SSST-ViT has better performance in terms of fault identification accuracy, loss value and stability.

The performance of the models was evaluated under the test set, and the accuracy and confusion matrix of different fault status identification models were obtained as shown in Table 6 and Figure 14, respectively.

It can be found in Table 6 and Figure 14 that the proposed method in this paper has the optimal diesel engine fault identification effect compared with other methods, and can effectively distinguish the confusing fault types.

In order to test the feature extraction ability of the SSST-ViT method, the output of the classification layer network of the ViT model was extracted as the discriminative features, and the results of fault status recognition were visualized in three dimensions by the t-SNE nonlinear dimensionality reduction technique, which is suitable for visualizing high-dimensional data. The original data of the training set, the original data of the test set, the feature data of the training set and the feature data of the test set were obtained, as shown in Figure 15.

In Figure 15, using the test set feature data as an example, since none of the methods proposed in this paper achieved 100% accuracy under the test set, there must have been some points that did not fall within a cluster. In other words, it is because some features are identified as features of other fault statuses that some feature points are not in a cluster and therefore the accuracy is not 100%. As can be seen in Figure 15, the SSST-ViT method has excellent feature extraction performance, and the features of each fault status in the space have obvious differentiability. Different fault status types are distributed in different locations in the space and exhibit dense clustering.

In summary, the effectiveness and superiority of the proposed diesel engine fault status recognition method are verified. The SSST-ViT method can effectively extract fault features and has high recognition accuracy compared with other methods.

## 4. Conclusions

In this paper, a diesel engine fault status identification method based on SSST and ViT is proposed with a diesel engine as the engineering application background. Compared with the traditional method, the following conclusions can be drawn:(1)SSST combines the high time-frequency aggregation of SST and the adaptive nature of ST, with better time-frequency aggregation and resolution.(2)The method is the first to apply SSST, which can effectively characterize the original signal features, and the ViT model, which has excellent image classification capability, to the field of diesel engine fault status identification, and can effectively extract time-frequency image features.(3)The method can provide theoretical and technical support for the research of diesel engine fault status recognition, which is of great military significance and realistic demand for improving the reliability and maintenance support capability of diesel engines.

Limited by the current experimental conditions, there are limitations in the development of diesel engine pre-set fault experiments. The focus of the next step is to carry out experimental research on diesel engine fault status identification in combination with diesel engine multi-part synthesis.

## Figures and Tables

**Figure 1 sensors-23-06447-f001:**
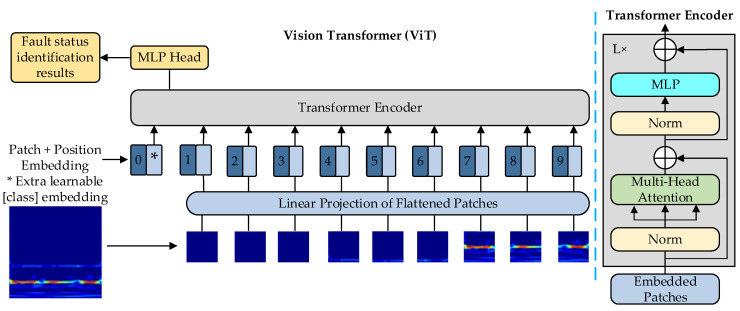
The architecture of the Vision Transformer.

**Figure 2 sensors-23-06447-f002:**
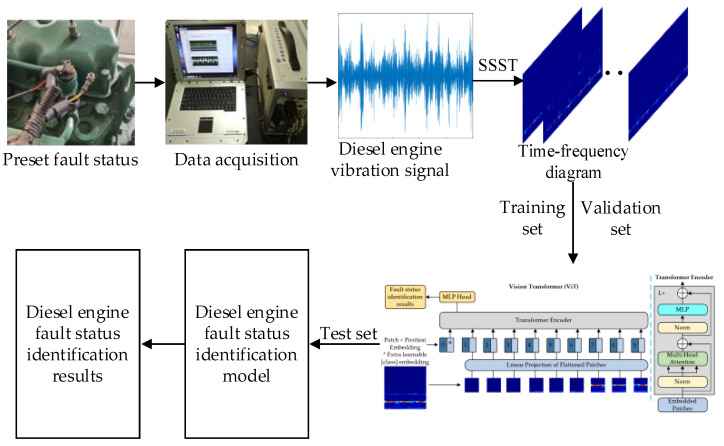
SSST-ViT fault status identification flow chart.

**Figure 3 sensors-23-06447-f003:**
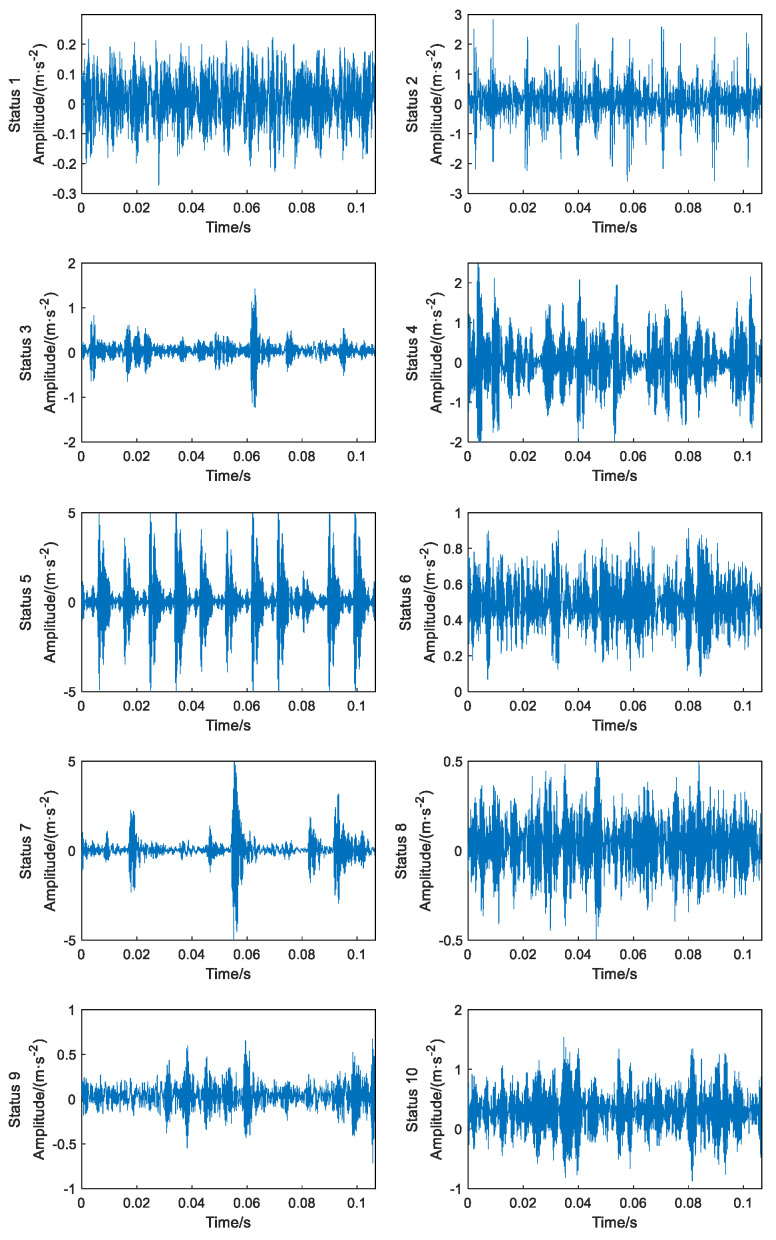
Time-domain waveforms of 10 failure statuses.

**Figure 4 sensors-23-06447-f004:**
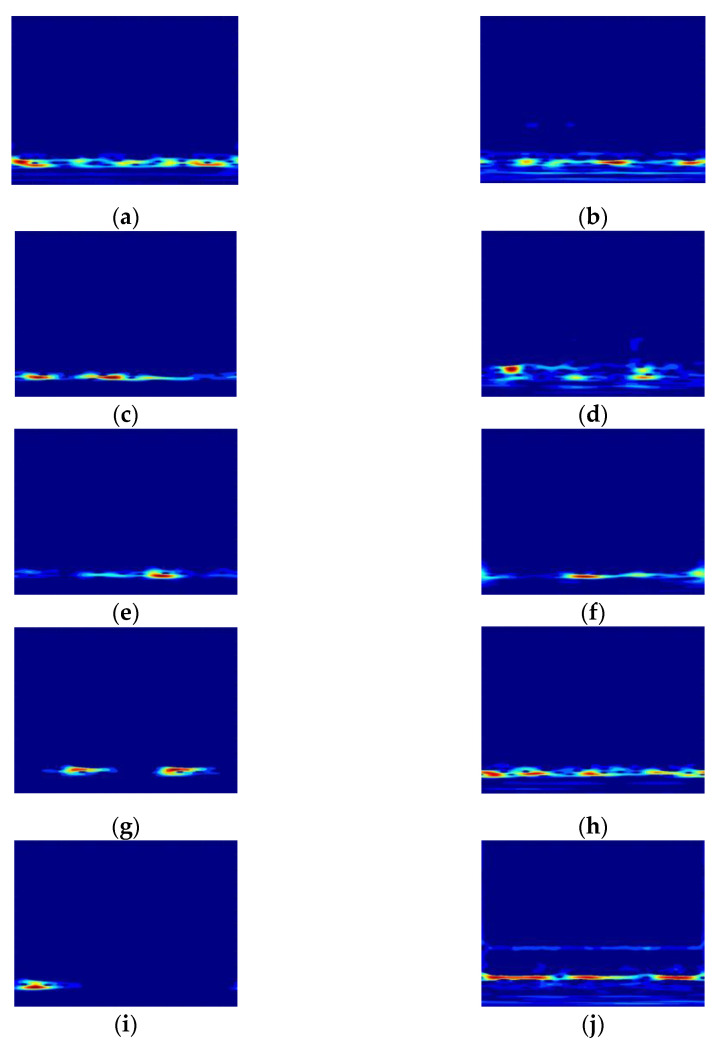
Time-frequency diagram of 10 fault statuses. (**a**) Status 1; (**b**) Status 2; (**c**) Status 3; (**d**) Status 4; (**e**) Status 5; (**f**) Status 6; (**g**) Status 7; (**h**) Status 8; (**i**) Status 9; and (**j**) Status 10.

**Figure 5 sensors-23-06447-f005:**
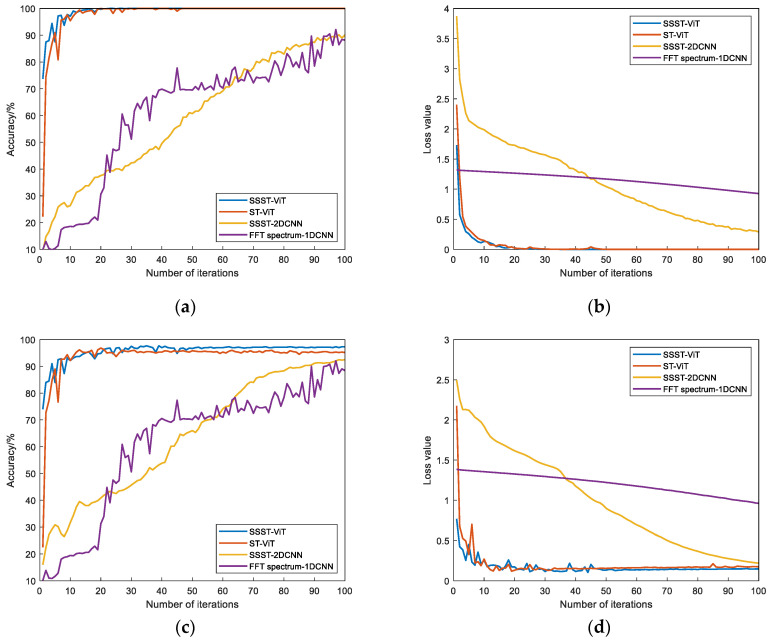
Comparison of training results of different models: (**a**) Accuracy of training set; (**b**) loss value of training set; (**c**) accuracy of validation set; and (**d**) loss value of validation set.

**Figure 6 sensors-23-06447-f006:**
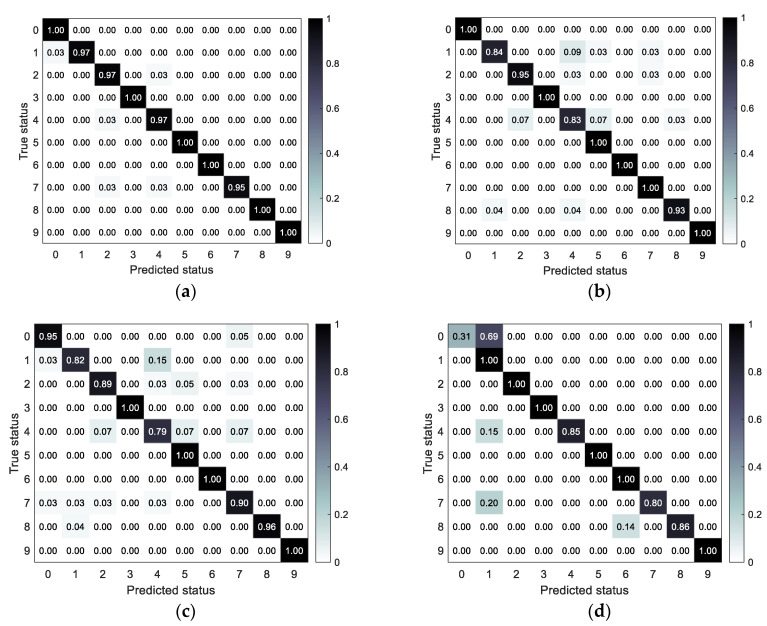
Confusion matrix for each model under the test set: (**a**) SSST-ViT; (**b**) ST-ViT; (**c**) SSST-2DCNN; and (**d**) FFT spectrum-1DCNN.

**Figure 7 sensors-23-06447-f007:**
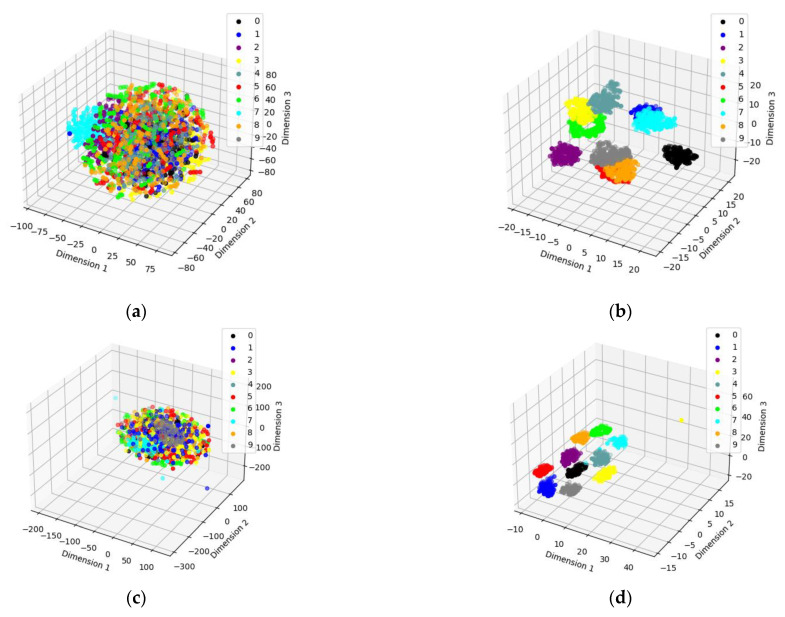
Three-dimensional visualization of status recognition results: (**a**) Training set raw data; (**b**) training set feature data; (**c**) test set raw data; and (**d**) test set feature data.

**Figure 8 sensors-23-06447-f008:**
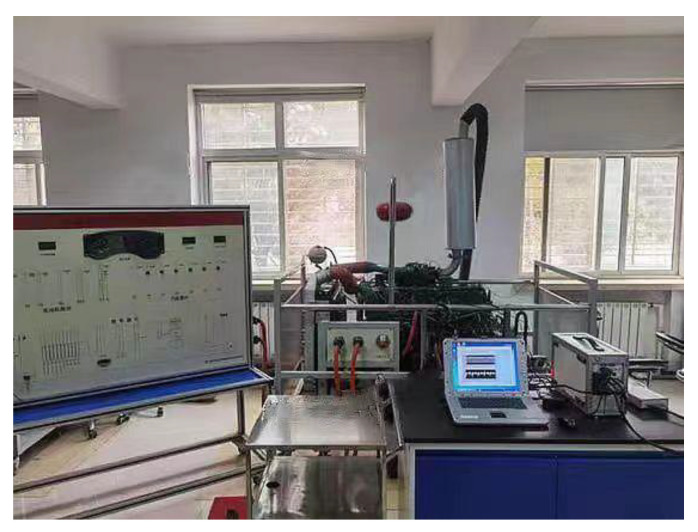
Diesel engine condition monitoring test bench.

**Figure 9 sensors-23-06447-f009:**
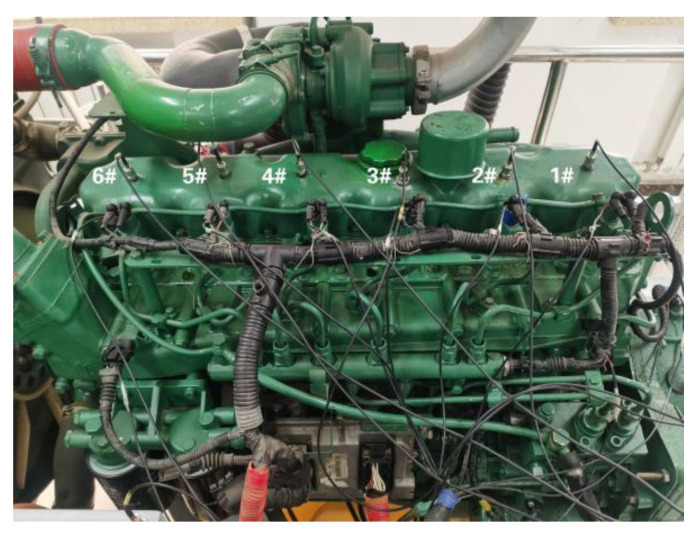
Sensor mounting position.

**Figure 10 sensors-23-06447-f010:**
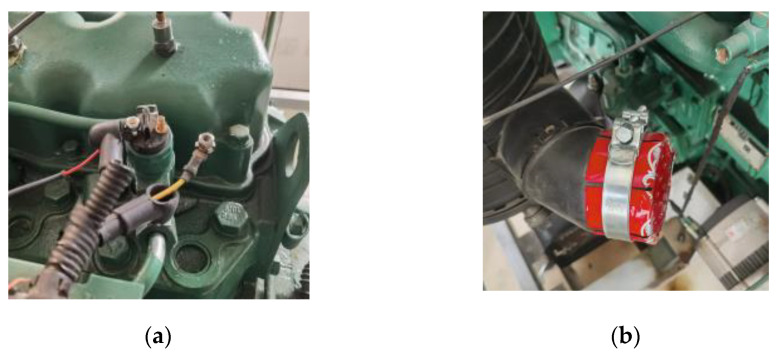
Pre-set diesel engine faults: (**a**) Disconnect the ignition power cord. (**b**) Install the air intake cover.

**Figure 11 sensors-23-06447-f011:**
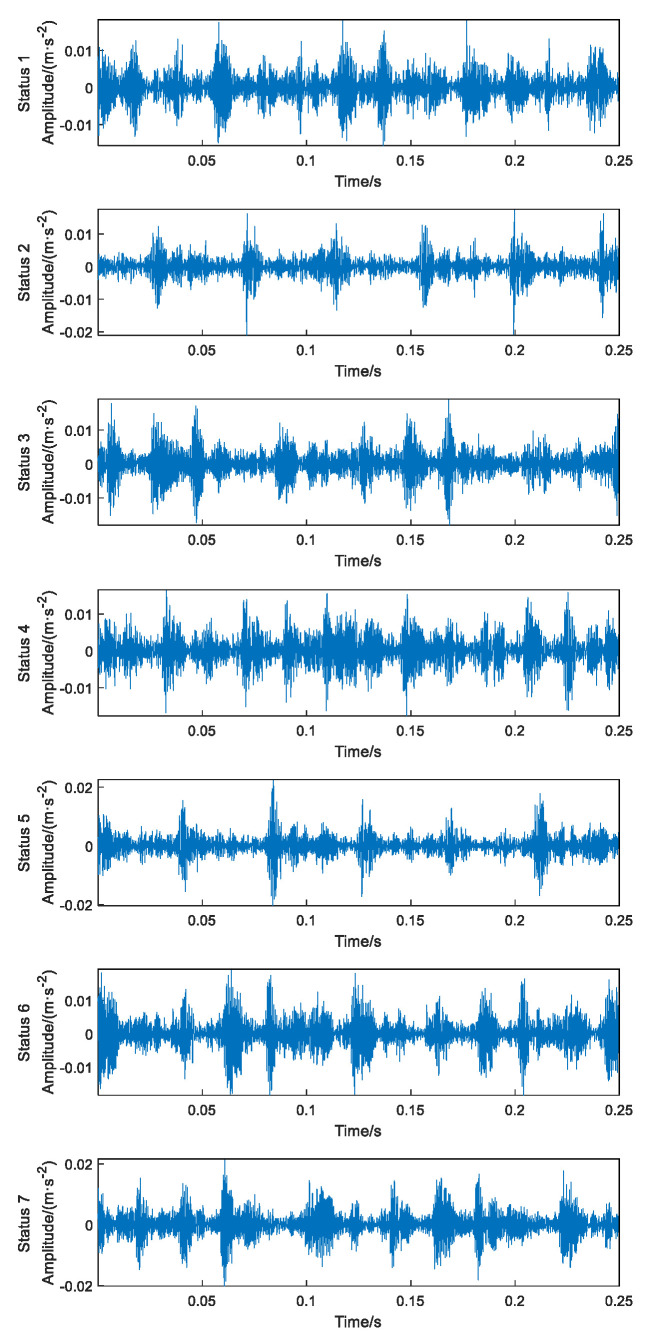
Time-domain wave forms of 7 failure statuses.

**Figure 12 sensors-23-06447-f012:**
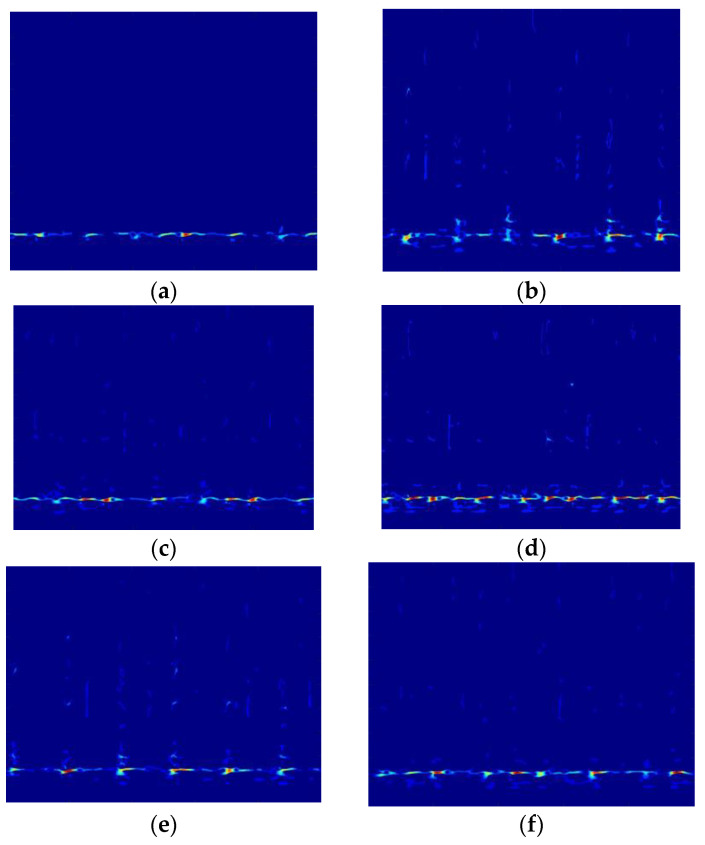
Time-frequency diagram of 7 fault status: (**a**) Status 1; (**b**) Status 2; (**c**) Status 3; (**d**) Status 4; (**e**) Status 5; (**f**) Status 6; and (**g**) Status 7.

**Figure 13 sensors-23-06447-f013:**
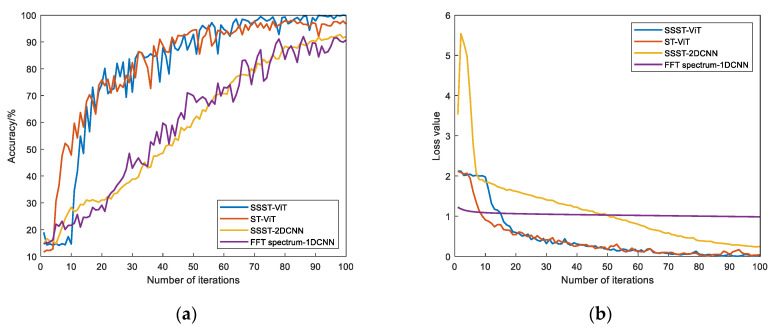
Comparison of the training results of each model: (**a**) Accuracy of training set; (**b**) loss value of training set; (**c**) accuracy of validation set; (**d**) loss value of validation set.

**Figure 14 sensors-23-06447-f014:**
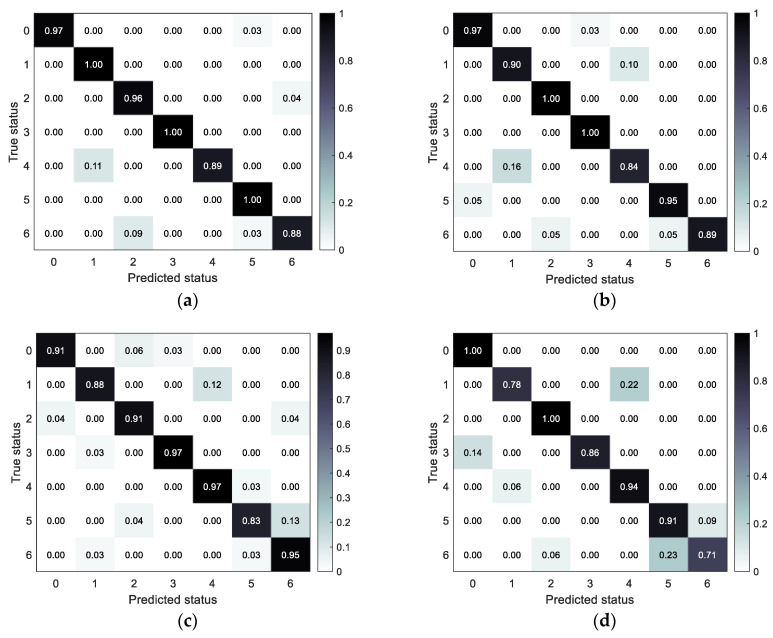
Confusion matrix for different fault status identification models: (**a**) SSST-ViT; (**b**) ST-ViT; (**c**) SSST-2DCNN; and (**d**) FFT spectrum-1DCNN.

**Figure 15 sensors-23-06447-f015:**
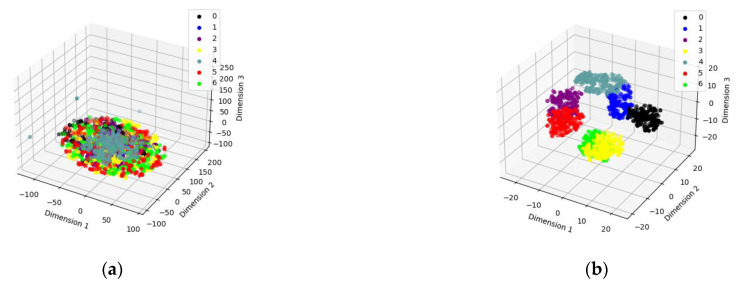
Three-dimensional visualization of status recognition results: (**a**) Training set raw data. (**b**) Training set feature data. (**c**) Test set raw data. (**d**) Test set feature data.

**Table 1 sensors-23-06447-t001:** The 10 kinds of bearing data selected.

Serial Number	Fault Location	Fault Diameter (mm)	Load (hp)	Rotational Speed (r/min)
1	Normal	—	0	1797
2	Inner ring failure	0.1778	0	1797
3	Inner ring failure	0.3556	0	1797
4	Inner ring failure	0.5443	0	1797
5	Outer ring failure	0.1778	0	1797
6	Outer ring failure	0.3556	0	1797
7	Outer ring failure	0.5443	0	1797
8	Rolling body failure	0.1778	0	1797
9	Rolling body failure	0.3556	0	1797
10	Rolling body failure	0.5443	0	1797

**Table 2 sensors-23-06447-t002:** Accuracy and loss values for each model.

Models	Accuracy/ (%)		Loss Value	
	Training Set	Validation Set	Training Set	Validation Set
Model 1	100.00	97.33	2.08 × 10^−4^	1.47 × 10^−1^
Model 2	100.00	95.17	2.27 × 10^−4^	1.74 × 10^−1^
Model 3	90.09	92.50	2.89 × 10^−1^	2.19 × 10^−1^
Model 4	88.17	88.40	9.27 × 10^−1^	9.62 × 10^−1^

**Table 3 sensors-23-06447-t003:** Accuracy of each model under the test set.

Models	Accuracy
SSST-ViT	98.31%
ST-ViT	95.27%
SSST-2DCNN	92.33%
FFT spectrum-1DCNN	88.50%

**Table 4 sensors-23-06447-t004:** Diesel engine preset fault mode.

Serial Number	Failure Mode
1	Normal
2	Fire in the first cylinder
3	Second cylinder fire
4	Clogged air filter
5	First cylinder and second cylinder misfire
6	Clogged air filter and first cylinder misfire
7	Clogged air filter and second cylinder misfire

**Table 5 sensors-23-06447-t005:** Accuracy and loss values for each model.

Models	Accuracy/ (%)		Loss Value	
	Training Set	Validation Set	Training Set	Validation Set
Model 1	99.86	95.43	6.46 × 10^−2^	1.69 × 10^−1^
Model 2	96.70	91.47	4.10 × 10^−2^	2.53 × 10^−1^
Model 3	92.00	93.33	2.48 × 10^−1^	2.25 × 10^−1^
Model 4	90.68	88.33	9.85 × 10^−1^	1.29

**Table 6 sensors-23-06447-t006:** Accuracy of each model under the test set.

Models	Accuracy
SSST-ViT	95.67%
ST-ViT	94.23%
SSST-2DCNN	91.90%
FFT spectrum-1DCNN	87.62%

## Data Availability

The authors confirm that the data supporting the findings of this study are available within the article.

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
