# Peer review of "Research on Diesel Engine Fault Status Identification Method Based on Synchro Squeezing S-Transform and Vision Transformer"

_sensors, 2023, doi:10.3390/s23146447_

Round 1

Reviewer 1 Report

·       The abstract can be restructured as follows: Research problem, research challenge/motivations (should be derived from literature), work done, and its justification, and key result indications.

·       There should be a sufficient number of keywords that give proper visibility to the paper.

·       Literature review should be improved. Currently, very few related papers are reviewed.   

·       Figures are not legible.

·       What are the disadvantages of SSST and ViT methods? Will it be accumulated in the proposed model?

·       How to handle the errors present in the sensor output?

·        Data link should be added to the paper.

·       From the results, ST-ViT also produces similar results compared to SSST-ViT. Is there any comparison available for the computation burden for each model?

·       Table 3 shows almost a 3 percent difference in accuracy between SSST-ViT and ST-ViT. However, the same is not clearly visible in Table 3.

·       Data available is one dimensional and that is converted in to two dimensional. What is the need of three-dimensional figures?

·       Conclusion should be improved.

Comments are added above. 

Reviewer 2 Report

This article a status identification method based on SSST - ViT is proposed, which can provide theoretical and technical support for the study of prognostics and health management of diesel engines. Firstly, the original signal is represented as a two-dimensional time-frequency map with high time-frequency aggregation by SSST, and then the time-frequency map is input to ViT as a feature map for training to realize the fault status recognition of diesel engines by combining the excellent performance of ViT model in the field of computer vision research. The proposed method is the first time to apply ViT network to the field of diesel  engine fault status recognition. Compared with traditional methods, the proposed  method has better recognition accuracy and stability, and it can effectively identify single  faults, mixed faults and other confusing fault types, and the results are better than other  methods.

 The choice of the proposed methods is justified. This approach is relevant. The research is of practical importance.

The conclusions correspond to the presented evidence and arguments, and they correspond to the basic question. The references appropriately. The tables, pictures and formulas meet the requirements.

1. Conduct research on the effectiveness of artificial intelligence methods for solving this problem - neural networks, classification, clustering, etc.

Reviewer 3 Report

The manuscript “Research on diesel engine fault status identification method based on synchro squeezing S-transform and vision transformer” applied the vision transformer deep learning method on the S-transformed fault status of diesel engines, to improve the accuracy of pattern recognition. The manuscript is clearly written overall, and the results show the proposed method (i.e., SSST-ViT) has higher accuracy and efficiency than a few other fault status recognition methods. I recommend publication after major revisions and clarifications:

1.     You compared the proposed method SSST-ViT with three models: ST-ViT, SSST-2DCNN, and FFT spectrum-1DCNN. Were those models proposed by other researchers, or did you develop them? I didn’t see references to the model names in the introduction. The proposed method should be compared with state-of-the-art methods that other researchers developed to show the improvements. Otherwise, comparing the proposed method with a few downgraded versions of the method doesn’t prove anything.

2.     What are the meaning and units of Figure 8 and Figure 16’s axes?

3.     In Figure 16 (d), due to a few “anomalies”, the clusters are not visually as spread as other plots such as Figure 16 (b) and Figure 8 (d). It would be clearer if you could make the axis’s range identical or similar to Figure 16 (b) and remove “anomalies”.

4.     Can you do similar t-SNE analyses for other methods you intend to compare with?

Round 2

Reviewer 1 Report

The authors have addressed most of the questions. However, the following question needs careful reconsideration.

·       From the results, ST-ViT also produces similar results compared to SSST-ViT. Is there any comparison available for the computation burden for each model?

1.       What is the significance of minor errors?  How does it impact various applications?

2.       Why computation burden is not considered as a performance parameter? 

Reviewer 3 Report

The authors failed to address any of my previous comments or questions. My previous comments and questions were intended to help the authors to improve the manuscript to make the information clear to readers, but the authors did not even try to improve the manuscript at all. The presentation of the approach and results have to be improved, otherwise, I cannot recommend publication. Here are my detailed responses to each of the points that the authors replied:

Point 1.

I have read the introduction so there is no need to copy and paste the lengthy text in your reply.  I was totally lost in your pasted texts and I tried really hard to find your main points that actually addressed my comments.

The only paragraph relevant to my comments in your reply is “The comparative methods in this paper are all models obtained with minor improvements based on the results of existing researchers and are all very representative. Therefore, it is reasonable and valid to demonstrate the superiority of the proposed methods through the comparison methods chosen in this paper.” If this is true, then my follow-up questions are:

1.     Based on which existing researchers’ results are Model 2: ST-ViT, Model 3: SSST-2DCNN, and Model 4: FFT spectrum-1DCNN improved, respectively? 

2.     Can you give references to each model? 

3.     Did you explicitly tell readers where these models come from, or how they are improved based on existing research, in your manuscript, when their names come up the first time? 

I completely understand your point that all these models you compared may be superior to traditional methods (and I trust you), but you CANNOT just claim it. Without showing or proving it, the research is incomplete and unconvincing.

Point 2.

Again, my comments and questions are for you to make the manuscript clearly presented, so please directly address them without copying and pasting your manuscript. I know those three axes are three dimensions, but in a scientific paper, these axes should have meanings. It’s not a toy example in an online tutorial. For example, can you at least let readers know which axes are dimension 1, dimension 2 and dimension 3, respectively? Can you label the axes in your plots?

Point 3.

I agree with you on this, but can you at least mention and briefly explain the phenomenon that some points are not within a cluster?  Now that you used t-SNE plots, then you need to explain to readers what can be clearly seen in the plots.

Point 4.

First of all, as mentioned in Point 1, you didn’t verify the superiority of the proposed method yet, because you haven’t shown a comparison with any existing methods. You haven’t explicitly told readers where those three models are coming from or derived from in your manuscript yet.

Secondly, if you don’t show t-SNE plots for other models, then what’s the point of saying SSST-ViT has an excellent feature extraction performance based on t-SNE in lines 497 - 499? How does it help to lead to the summary in lines 501 – 503, where you mentioned the high recognition accuracy? 

Besides, I don't understand why you have to show 16 images. Only showing the t-SNE plots for test feature data would suffice if SSST-ViT is superior in terms of feature extraction. That’s only 4 images in total.
